# Differences in Pathogenesis-Related Protein Expression and Polyphenolic Compound Accumulation Reveal Insights into Tomato–*Pythium aphanidermatum* Interaction

**Seham A. Soliman [1], Abdulaziz A. Al-Askar [2], Sherien Sobhy [1], Marwa A. Samy [1], Esraa Hamdy [1], Omaima A. Sharaf [3], Yiming Su [4], Said I. Behiry [5],\* and Ahmed Abdelkhalek [1],\***

[1] Plant Protection and Biomolecular Diagnosis Department, Arid Lands Cultivation Research Institute, City of Scientific Research and Technological Applications, Alexandria 21934, Egypt

[2] Department of Botany and Microbiology, College of Science, King Saud University, P.O. Box 2455, Riyadh 11451, Saudi Arabia

[3] Department of Agricultural Microbiology, National Research Centre, Cairo 12622, Egypt

[4] Utah Water Research Laboratory, Department of Civil and Environmental Engineering, Utah State University, Logan, UT 84341, USA

[5] Agricultural Botany Department, Faculty of Agriculture (Saba Basha), Alexandria University, Alexandria 21531, Egypt

\* Correspondence: said.behiry@alexu.edu.eg (S.I.B.); aabdelkhalek@srtacity.sci.eg (A.A.); Tel.: +20-1007556883 (A.A.)

**Abstract:** Plant diseases significantly reduce crop yields, threatening food security and agricultural sustainability. Fungi are the most destructive type of phytopathogen, and they are responsible for major yield losses in some of the most crucial crops grown across the world. In this study, a fungus isolate was detected from infected tomato plants and molecularly identified as *Pythium aphanidermatum* (GenBank accession number MW725032). This fungus caused damping-off disease and was shown to be pathogenic. Moreover, the expression of five pathogenesis-related genes, namely *PR-1*, *PR-2*, *PR-3*, *PR-4*, and *PR-5*, was quantitatively evaluated under the inoculation of tomato with *P. aphanidermatum*. The quantitative polymerase chain reaction (qPCR) showed that the expression levels of *PR-1*, *PR-2*, and *PR-5* genes went up significantly at 5 days post-inoculation (dpi). The expression of the *PR-1* gene also increased the variably, which reached its highest value at 20 dpi, with a reported relative expression level 6.34-fold higher than that of the control. At 15 dpi, *PR-2* and *PR-5* increased the most, while *PR-1*, *PR-3*, and *PR-5* also increased noticeably at 20 dpi. On the contrary, *PR-4* gene expression significantly decreased after inoculation, at all time intervals. Regarding *PR-5* gene expression, the data showed a variable change in *PR-5* gene expression at a different sample collection period. Still, it was highly expressed at 15 dpi and reached 3.99-fold, followed by 20 dpi, where the increasing percentage reached 3.70-fold, relative to the untreated control. The HPLC analysis indicated that the total concentration of all detected polyphenolic compounds was 3858 µg/g and 3202.2 µg/g in control and infected plant leaves, respectively. Moreover, the HPLC results concluded that *Pythium* infection decreased phenolic acids, such as chlorogenic and ellagic acids, which correlated with the infection–plant complex process. Based on the results, *P. aphanidermatum* could be a biotic stress pathogen that causes the expression of pathogen-related genes and stops the regulation of defensin phenolic compounds.

**Keywords:** agricultural sustainability; *Pythium aphanidermatum*; tomato; PR genes; damping-off; HPLC; gene expression

## 1. Introduction

Every year, plant diseases cause massive economic crop losses all over the world [1]. The diseases caused by fungi are the most common, and they have been a persistent threat to food and feed security since the domestication of agricultural plants [2]. The use of

synthetic fungicides is the most common method used to prevent and treat diseases caused by fungi. The overuse of pesticides harms land, water, and humans. This threatens the sustainability of agriculture by harming farmers, consumers, and agricultural resources. Among fungal pathogens, *Pythium* spp. produce pectinases and cellulases to partially puree cellular structure in order to gain entry to simple sugars inside of plant tissues [3]; thus, it is called "sugar fungi" because it degrades simple carbohydrate polymers [4] and appears to rely on them for growth. Root rot and damping-off diseases caused by *Pythium aphanidermatum* are considered the most prominent diseases in all tomato-growing areas [5].

Tomato (*Solanum lycopersicum* L.) is one of the essential vegetable crops globally, and it can be used in diverse ways, including fresh as well as processed [6]. In Egypt, tomato accounts for about 16% of cumulative vegetable production, despite only occupying about 32% of the total vegetable growing area [7]. Tomato production is facing a threat from multiple fungal diseases. It is worth noting that plants employ various immune mechanisms, which begin with recognizing pathogens, triggering defense signal pathways, and generating antifungal substances, such as pathogenesis-related (PR) proteins [8]. These proteins help prevent pathogen infiltration and reduce its growth. [9]. The pathogenesis-related (PR) proteins consist of different molecules stimulated by both phytopathogens and signaling molecules related to defense. The PRs are widely used as diagnostic molecular markers for defense signaling pathways because they comprise the bulk of the plant's innate immune system [10]. PR genes have been shown to increase resistance to both biotic and abiotic stresses, making them a promising strategy for creating crop varieties that can withstand a wide range of environmental stresses [11].

It has been reported that many polyphenolic, phenolic, and flavonoid compounds in herbal-originated extracts possess higher anti-oxidative activity than those of vitamins C and E. Their anti-oxidant properties stop reactive oxygen species from being cleaned up or indirectly chelating transition metals [12]. In plants, it is well known that polyphenolics play a significant role in the plant's ability to protect itself from both biotic and abiotic stresses [13–15]. Plants make phenolic compounds, including polyphenols, flavonoids, anthocyanins, phenolic acids, and phenolic terpenes, to control important physiological processes and protect themselves from biotic stress, such as oxidative stress caused by pathogens [16]. Often, these phenolic compounds build up at the site of an infection to slow the growth of invading microorganisms and stop them from spreading through the plant tissue [17]. Cheynier et al. [18] suggest that this is achieved by increasing the number of reactive species and free radicals.

From a scientific point of view, studying how the *Pythium* pathogen affects the tomato can help us learn more about how plants and pathogens interact and how plants fight off pathogens. This can lead to the creation of new technologies and ideas that can help keep plant diseases under sustainable control. In the case of *P. aphanidermatum* infecting tomato plants, the expression of several PR genes involved in plant defense was examined to determine how well plants and pathogens get along. This analysis was performed during the early stages of infection, specifically at 5, 10, 15, and 20 days post-inoculation. HPLC analysis also showed that Pythium infection caused changes in the plant's metabolic activity and the number of polyphenolics.

## 2. Materials and Methods

### 2.1. Sample Collection

The tomato plant samples were collected from the local open fields at Borg EL Arab city in Egypt's Alexandria Governorate. Tomato plants that suffer from these symptoms (chlorosis, wilting, damping-off, and root rot) were harvested and transported to the laboratory in sterilized plastic bags.

### 2.2. Isolation and Purification of the Pathogen

Once at the laboratory, the affected tomato plants were carefully separated into their root and aerial parts using a sterilized scalpel. The sample parts were washed gently with

tap water and scissored into small pieces measuring about 0.5–1.0 cm. These pieces were then superficially sterilized for 1 min in a 0.5% NaOCl (*w/v*) solution before being washed several times with sterile double distilled water (SDDW) and dried. The sterilized small pieces were then cultured on the surface of potato dextrose agar (PDA) plates for 7 days at $28 \pm 2$ °C. To purify the fungus isolate, the hyphal tip technique [19] was employed, and each fungal colony was placed on fresh PDA plates for 7 days at $28 \pm 2$ °C to confirm its purity. The cleaned fungal isolates were then stored on PDA slants until they were used in other experiments.

### 2.3. Fungus Characterization

2.3.1. Morphological Identification

The primary identification of each isolate was performed by analyzing its morphological and cultural characteristics [20,21]. The morphological identification of the fungus isolate was performed using a lactophenol cotton blue solution stain. The stain was applied to the slide, followed by the transfer of the mycelia growth; the stain and mycelia were then covered by a slip. Finally, the prepared slide was observed under the microscope [22].

2.3.2. Molecular Identification of Fungi Using PCR

DNA Isolation of Fungal Cells

The method used to isolate the DNA from the fungus was based on that of Tiwari et al. [23]. The crushed fungus mycelium was treated with 1.5 mL of cetyltrimethylammonium bromide (CTAB) buffer before being placed in an Eppendorf tube with 10 µL of beta-mercaptoethanol. The mixture was vortexed and held at 65–70 °C for half an hour. Next, the tube mixture was allowed to spin at 12,000 rpm for 10 min, the supernatant lysate was moved to a new tube, and the same amount of phenol was added. The tube was spun at 12,000 rpm, the top layer was transferred to a fresh tube, and the same volume of chloroform was added to eliminate the phenol residue. DNA was precipitated by combining the sample with isopropyl alcohol and freezing it at $-80$ °C for 1 h before centrifuging it at 12,000 rpm for 20 min. The aqueous layer was discarded, and the pellet was washed with 750 µL of 75% ethanol before centrifuging it at 12,000 rpm for 10 min. The pellet was resuspended in 70 µL of SDDW following the removal of the supernatant, and the DNA was subsequently analyzed by gel electrophoresis and quantification.

Agarose Gel Electrophoresis

The fungus' genomic DNA was analyzed using a 1% agarose gel prepared in tris-borate EDTA (0.5X TBE), as described by Sambrook et al. [24], with the gel being prepared in a microwave. The agarose gel was loaded onto an electrophoresis apparatus in TBE 0.5X, and the DNA was visualized on an ultraviolet transilluminator (UV) after staining with ethidium bromide. The gel was photographed using a gel documentation system.

Detection of the Fungus by ITS-PCR

A primer was utilized to detect and amplify the rDNA-ITS region of the fungal isolate. The PCR reaction mixture consisted of 10 µL of master mix, 1 µL of DNA (30 ng), universal primer ITS (forward TCC GTA GGT GAA CCT GCG G and reverse TCC TCC GCT TAT TGA TAT G), 1 µL for each primer (10 pmol/µL), and sterile $dH_2O$, up to a final volume of 20 µL. The PCR program involved initial denaturation at 95 °C for 3 min, followed by 35 cycles of 94 °C for 1 min, annealing at 50 °C for 1 min, and extension at 72 °C for 1 min. The final extension step was performed at 72 °C for 5 min. Subsequently, 5 µL of the PCR products were separated on 2% (*w/v*) agarose gel electrophoresis, and the gel was photographed using a gel documentation system.

PCR Product Purification

A PCR clean-up column kit (Koma Biotech, Seoul, Republic of Korea) was used to purify the amplified PCR products, according to the manufacturing procedures. The DNA

fragment was excised from the gel and weighed in a 1.5-mL Eppendorf tube. For each 100 mg of gel weight, 400 µL of binding buffer II was added and incubated in a water bath at 65 °C until it dissolved. The mixture was transferred to the EZ-10 column and allowed to stand at room temperature for 2 min before being centrifuged at 10,000 rpm for 2 min, and the flow-through was discarded. In a repeated step, 750 µL of wash solution was added to the column and then spun for 1 min at 10,000 rpm. The free columns were spun at 10,000 rpm for 1 min to remove the buffer residues. The columns were placed in a clean 1.5 mL Eppendorf tube, and 50 µL elution buffer was added to the column center and left for 2 min before being spun at 10,000 rpm for 2 min to elute DNA. The collected DNA was stored at −20 °C until use.

DNA Sequencing and Phylogenetic Construction

The sequence was submitted to the NCBI GenBank database, the DNA sequence was aligned with other organism strains of the same species available in the GeneBank database, and the phylogenetic tree was constructed with MEGA 11 software [25].

*2.4. Greenhouse Experiment*

To estimate the effect of *Pythium* spp. on tomato plants, a greenhouse experiment was carried out. For a detailed pathogenicity experiment, *Pythium* inocula were prepared at a concentration of 5%. To prepare the inoculated soil, 5 g of grass blade (*Echinochloa colonum*) leaf segments measuring $1 \times 0.5$ cm and 2 g of glucose were mixed with 10 mL of SDDW in an Erlenmeyer flask. Each flask was autoclaved at 121 °C for 20 min and inoculated with three 5 mm diameter discs of *Pythium* mycelium cultured on PDA medium. The inoculated flasks were incubated at 28 °C for 10 days. A 5% inoculum concentration was achieved by grinding 2 g of colonized grass leaf segments with 50 g of sterilized clay–sand soil (1:1 *w/w*). In a plastic bag, 5 g of the inoculated soil were combined with 95 g of moistened sterilized clay–sand soil and incubated at 28 °C for 2 weeks before use. The experiment was distributed among five replicate pots (500 g of clay-sand mixture soil), and the inocula were added to the soil after the seedlings were planted in each pot. The tomato seedlings not inoculated with *Pythium* served as the controls.

*2.5. Sample Collection and RNA Isolation*

Fungal-inoculated tomato plant leaf samples were collected 5, 10, 15, and 20 days post-inoculation (dpi). The samples were weighted and kept at −80 °C for RNA isolation. Total RNA was extracted from the samples, according to the methods of Chomczynski and Sacchi [26]. Frozen tissues were ground in liquid nitrogen immediately after dissection. A total of 1 mL of denaturing solution (4 M guanidinium thiocyanate, 25 mM sodium citrate, pH 7.0, 0.1 M 2-mercaptoethanol, and 0.5% (*w/v*) N-laurosyl-sarcosine) was added to 100 mg of pulverized tissue and transferred to a new Eppendorf tube. Then, 0.1 mL of 2 M sodium acetate, pH 4.0, was added to 1 mL of lysate and inverted 2–3 times; after that, 1 mL of water-saturated phenol was added and inverted again; then, 0.2 mL of chloroform/isoamyl alcohol (49:1) was added. The mixture was mixed by hand for 10 s, put on ice for 15 min, and centrifuged for 20 min at $10,000\times g$ at 4 °C. The aqueous layer was transferred to the new Eppendorf tube, and 1 mL of isopropanol was added to precipitate the RNA. The mixture was incubated at −80 °C for 1 h and centrifuged at $10,000\times g$ for 20 min at 4 °C; the supernatant was discarded. The RNA pellet was resuspended in 750 µL of 75% ethanol and vortexed for a few seconds, centrifuged at $10,000\times g$ for 5 min at 4 °C, and the supernatant was discarded. The RNA pellet was air-dried for 5–10 min at room temperature. The RNA concentration was measured using a Nanodrop 2000c spectrophotometer (Thermo, Waltham, MA, USA). The purity of the RNA was calculated, and it was stored at −80 °C until use.

*2.6. Reverse Transcription–Polymerase Chain Reaction (RT-PCR)*

A process was carried out to create complementary DNA (cDNA) by the reverse transcription of total RNA. The reaction mixture contained 3 μL of total RNA, 5 μL of oligo (-dT) primer (10 pmol), 2.5 μL of dNTPs, 2.5 μL of buffer (10×), 0.3 μL of reverse transcriptase (M-MULV, Biolabs, London, UK), and up to 20 μL of sterile dH$_2$O. The reaction mixture was gently mixed by shaking and subjected to amplification, which was programmed as follows: 42 °C for 1.5 h, followed by inactivation at 80 °C for 10 min in a thermocycler (MJ Research, Inc., PTC-100TM Programmable Thermal Controller, Deltona, FL, USA). The resulting cDNA-RNA was then stored at −20 °C.

*2.7. Real-Time PCR*

The qRT-PCR was performed in triplicate using SYBR Green master mix in 25 μL total reaction volume. Primers for five specific defense genes were used in this study, as shown in (Table 1). The reaction mixture contained: 12.5 μL SYBR Green, 1 μL of 10 pmol forward primer, 1 μL of 10 pmol reverse primer, and 1 μL of cDNA (50 ng), and the volume was completed up to 25 μL with sterile dH$_2$O. The samples were spun in the centrifuge before running. The reaction conditions were as follows: initial denaturation at 95 °C for 15 min; 45 cycles of 95 °C for 10 s; annealing at 60 °C for 15 s; elongation at 72 °C for 30 s. Data acquisition was performed during the extension step. This reaction analysis was performed by Rotor-Gene-6000-system analysis (Qiagen, Germantown, MD, USA). The threshold of the cycle equal to each identified gene was determined by automated threshold analysis on the ABI System. The equations for the quantity gene expression were calculated according to the $2^{-\Delta\Delta CT}$ algorithm [27].

*2.8. HPLC Analysis*

Using a modified version of the technique provided by Ćetković et al. [28], 20 μL of the extract was injected onto a 0.45 μm nylon membrane for HPLC resolution with the use of an automated injector and an automatic degassing system from a Waters 2690 diode array detector. At a flow rate of 1 mL/min, a 4.6 mm inner diameter by 250 mm long Zorbax SB-C18 column packed with 5 μm particles size was utilized, with a mobile-phase combination of 1% formic acid in water *v/v* (A) and 100% methanol (B). The procedure involved a step gradient from 20% (B) and 80% (A) at 0–40 min to 65% (B) and 35% (A) at 40 min, 92% (B) and 8% (A) at 45 min, and 100% (B) and 0% (A) at 50 min. At last, the column was restored to working order. Absorption was measured at 254 and 324 nm. The chemicals were recognized by comparing the information included in their UV spectra to those found in a library of known standards. The phenolic content reported as μg/g was determined using standard curves.

*2.9. Statistical Analysis*

Statistical analysis was performed using the analysis of variance technique and the CoStat software v.6.311(CoHort software, Monterey, CA, USA). The attained experimental result was statistically analyzed according to the method of Gomez and Gomez [29]. The PRs expression levels were presented as means ± S.E., and statistical significance was considered when the *p*-value was less than or equal to 0.05, using Duncan's multiple range test [30].

**Table 1.** Nucleotide sequences of qPCR primers used in this study.

| Gene | Abbreviation | Sequence 5′——3′ | Function | References |
|---|---|---|---|---|
| Beta-actin | *β-actin* | Frw-ATGCCATTCTCCGTCTTGACTTG<br>Rev-GAACCTAAGCCACGATACCA | Housekeeping gene used for normalizing gene expression levels in many different types of cells. | [31] |
| *Pathogenesis-related protein-1* | *PR-1* | Frw-TTCTTCCCTCGAAAGCTCAA<br>Rev-CGCTACCCCAGGCTAAGTTT | Plays a role in the plant's defense against pathogens. PR-1 can bind to fungal cell walls and inhibit fungal growth; considered as a biomarker for plant stress response. | [32] |
| *β*-1,3-glucanases | *PR-2* | Frw-TCACCAAACTATTGGATTTCAA<br>Rev-GACTCAATTTTTGACTTCTTAATCC | Encodes the production of *β*-1,3-glucanase enzyme, which plays a role in defense against fungal pathogens. | [33] |
| Chitinase | *PR-3* | Frw-ACTGGAGGATGGGCTTCAGCA<br>Rev-TGGATGGGGCCTCGTCCGAA | Encoding the production of chitinase, the enzyme that breaks down chitin, found in the cell walls of fungi (cell wall degrading enzyme). | |
| Chitin binding protein | *PR-4* | Frw-GACAACAATGCGGTCGTCAAGG<br>Rev-AGCATGTTTCTGGAATCAGGCTG | Encodes a protein called chitinase-binding protein in plants, which binds to chitin and prevents fungal growth. | [34] |
| Thaumatin-like protein | *PR-5* | Frw-ATGGGGTAAACCACCAAACA<br>Rev-GTTAGTTGGGCCGAAAGACA | Thaumatin-like proteins (TLPs) are part of the family of PR proteins. Plants make more of these proteins when they are stressed by both biotic and abiotic factors. The *PR-5* gene encodes for a type of TLP in plants, and it may act by disrupting the integrity of the cell walls of pathogens. | [32] |

## 3. Results and Discussion

In the current study, research into the interaction between the *Pythium* pathogen and the tomato plant can contribute to sustainability by addressing challenges related to sustainable agriculture. By understanding the mechanisms of the Tomato–*Pythium* interaction, researchers can develop strategies to manage plant diseases in a sustainable manner, minimizing the negative impacts on agricultural productivity, human health, and the environment.

### 3.1. Isolation and Identification of the Fungus Isolate

In our study, tomato samples exhibited damping-off symptoms, which led us to isolate a pathogen from the infected tissues and purify it on PDA media (Figure 1A). Morphological analysis revealed that the fungus had branched, filamentous hyphae that were generally non-septate, lacking cross-walls, as well as spherically or pear-shaped oogonia that contain female gametes. Upon fertilization, the oogonia produce thick-walled oospores [20,21]. We identified the fungus using lactophenol cotton blue solution stain under a light microscope, and Figure 1B,C show the hyphae of *Pythium* spp., as well as the oogonia, oospores, and antheridia.

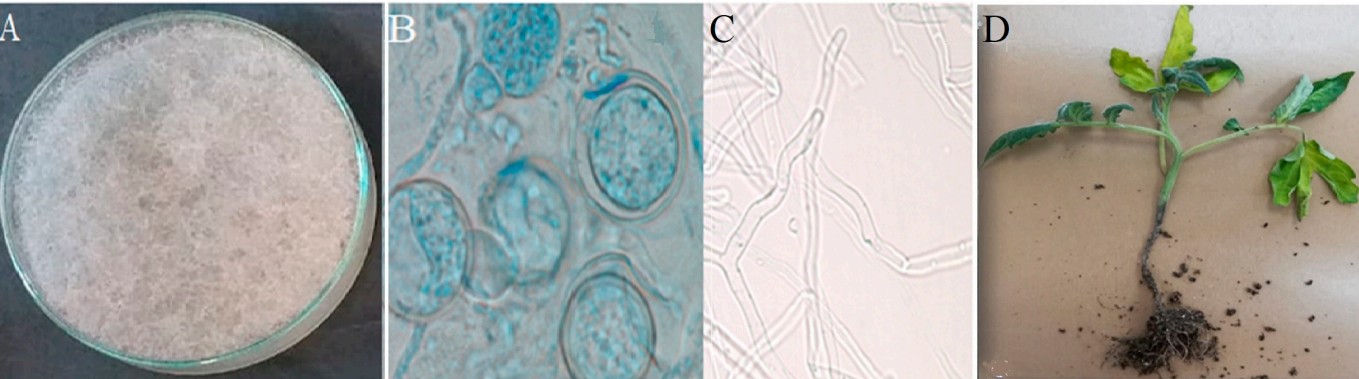

**Figure 1.** (**A**) Morphological mycelial growth of *Pythium aphanidermatum* grown on a PDA plate; (**B**) light microscope images showing oospores, oogonia, and antheridia; (**C**) *P. aphanidermatum* hyphae; (**D**) an infected tomato plant.

The internal transcribed spacer (ITS) region of fungal ribosomal DNA has gained popularity as a molecular marker for fungal identification, systematics, and phylogenetics. This is mainly due to its high sequence variability within and between fungal species, its conserved flanking regions, and the availability of a large number of reference sequences in public databases. Consequently, ITS has become known as a "fungal barcode" and is widely used for various fungal research applications [35]. The genomic DNA of the fungus isolate was subjected to PCR amplification using the ITS region; by comparing the PCR product and DNA marker, the size of the PCR product was 550–600 bp, as shown in Figure 2A.

The DNA nucleotide sequence of the fungus ITS region was produced at Macrogen Company (Seoul, Republic of Korea). After that, the nucleotide sequence was compared with the global recorded database in the NCBI, using the BLAST tool. The highest similarity of the fungus isolate determined it to be *P. aphanidermatum*. The fungus isolate nucleotide was then submitted to the GenBank database submission portal and assigned to the accession number (MW725032). The phylogenetic tree of the obtained fungus nucleotide (ITS) was constructed based on *Pythium*-fungal sequences obtained from the GenBank database, closely related to *P. aphanidermatum*. The constructed tree assigned all the *Pythium* species to two clusters. *P. aphanidermatum* isolates were divided into two sub-climbs. Subclimb two was split into two groups; group two included *P. aphanidermatum*, similar to the investigated isolate with different percentages (Figure 2B). In the same manner, a study conducted by Matsumoto et al. [36] revealed that the ITS region was a suitable target

for designing primers specific to *Pythium* species, which enabled their identification and detection. Analysis of the 5.8S rDNA sequence tree indicated that oomycetes, including *Pythium*, are evolutionarily distant from other fungi. The findings of Barboza et al. [37] also agree with our results of the phylogenetic tree, which grouped the same species in one clade of BLASTn ITS regions.

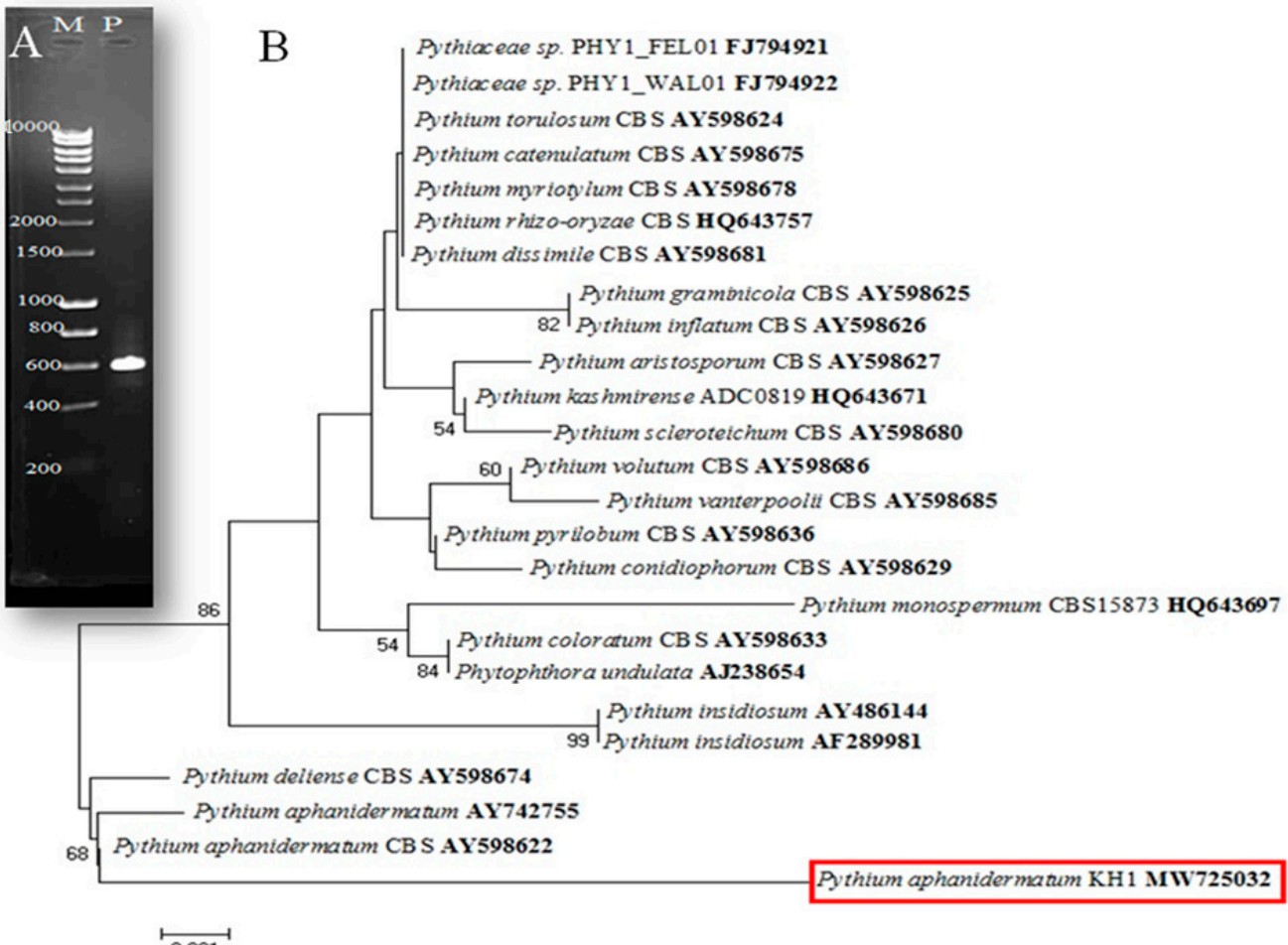

**Figure 2.** (**A**) PCR product of the ITS region of the fungus [M, 10 kb DNA marker, Lane 1 (P), *Pythium aphanidermatum* KH1 (MW725032)]; (**B**) phylogenetic tree of the ITS region of the *P. aphanidermatum* nucleotide sequence compared with other GenBank sequences.

### 3.2. Greenhouse Trail and Pathogenicity Test

This study sheds some light on changes during inoculation between the tomato and the necrotrophic fungal pathogen *Pythium aphanidermatum*. The obtained results revealed that *P. aphanidermatum* was implicated as a pathogen in tomato plants (Figure 1D). *P. aphanidermatum* causes damping-off disease symptoms in the inoculated tomato plant 25 days post-inoculation. The symptoms that appeared include leaf drop and wilting, and this finding may be attributed to the cell wall-degrading enzymes produced by the fungus, which soften the plant cell wall [4]. Moreover, the results also showed chlorotic leaves and stunted tomato growth. This result may be accredited to the need of this fungus for carbohydrates, as it depends on them for growth, disturbing metabolism, and enzyme action. The infection begins at the root tip and can cause the infected region to lose its protective outer layer [4]. Our findings concur with those of Muthukumar et al. [38], who evaluated the pathogenicity of several *Pythium* isolates, including *P. aphanidermatum*, *P. deliense*, *P. graminicola*, *P. heterothallicum*, and *P. ultimum* in chili plants. The results indicated that *P. aphanidermatum* was the most virulent isolate, causing a significant level of post-emergence damping-off (65%).

According to the findings of Garibaldi et al. [39], pathogenicity tests were carried out to evaluate the impact of *P. aphanidermatum* on *Beta vulgaris* subsp. *vulgaris* at two different temperatures: 27 °C and 20 °C. The results demonstrated that after a period of 20 days, a significantly higher percentage of plants exhibited damping-off symptoms at 27 °C (70%) in comparison to those seen at 20 °C (10%).

### 3.3. The Expression Levels of PR-Related Genes

Real-time PCR was used to detect the relative amounts of mRNA for five PR-related protein genes in tomato leaves infected with *P. aphanidermatum*. The results were normalized to the *β*-actin gene, as a reference or housekeeping gene. Among PR proteins, *PR-1*, *PR-2*, *PR-3*, *PR-4*, and *PR-5* have been evaluated as the most potent antifungal proteins in plants. The over-enhancement of these genes in several crops leads to enhanced disease resistance against biotrophic and necrotrophic fungal phytopathogens [40].

In the present study, the expression of the *PR-1* gene increased variably and reached its highest value at 20 dpi, with a reported relative expression level 6.34-fold higher than that of the control (Figure 3). Larsen et al. [41] reported similar findings, indicating that upon inoculation with *P. aphanidermatum*, tomato plants exhibited a high expression of the *PR-1* gene. The pronounced rise in *PR-1* gene expression relative to that of the uninoculated control was consistent with the findings of Moghaddam et al. [42], where up-regulation of the *PR-1* gene, encoding pathogenesis-related protein 1, was strongly elevated in tomato plants as a result of *Alternaria alternata* infection, supporting the possible involvement of these PR genes in the regulation and response to early fungal infection. Early research suggested that *PR-1* proteins have antimicrobial activity, with *PR-1* derived from *Nicotiana tabacum* being able to inhibit the growth of *Phytophthora infestans* zoospores [43]; this role may be due to *PR-1*'s ability to sequester sterols from pathogen membranes, resulting in growth inhibition [44]. Accordingly, the increase in *PR-1* expression is frequently used as an indicator for resistance responses, including pattern-associated triggered immunity (PTI) and systemic acquired resistance (SAR) [45].

Moreover, data also showed that *PR-2* gene (β-1,3-glucanase) expression was modified slightly at 5, 15, and 20 dpi, while it decreased significantly at 10 dpi relative to the control (Figure 3). Glucanases, abundant proteins in plants, play chief roles in many physiological processes, such as cell division, transferring materials through plasmodesmata, flower formation through seed maturation, and amending abiotic stress effects [46]. They also defend plants against fungal pathogens, alone or in combination with many genes, such as thaumatin-like proteins, peroxidases, and a-1-puro thionin, where they develop strong resistance against fungal pathogens in crop plants [46]. The obtained results were in agreement with many previous reports that showed the induction of *PR-2* upon plant fungal infections [47–49]. Thus, it is assumed that the up-regulation of tomato *PR-2* in the current study could be related to the defense response activity of tomato plants against *P. aphanidermatum* infection.

Chitinases, also known as *PR-3*, are a group of proteins that protect plants against fungal diseases by cleaving the bonds between chitin molecules that are found in the fungal cell wall [50,51]. In the present study, compared to the uninfected control, a remarkable rise in *PR-3*, with a relative expression level of 5.76-fold at 20 dpi was reported (Figure 3). Many researchers found that *A. alternata* and *A. solani* infections caused *PR-2* and *PR-3* expression to increase, showing that they protect against early blight disease [52,53]. Moreover, Jongedijk et al. [54] and Wubben et al. [55] evaluated the way in which inoculating tomato plants with *Fusarium oxysporum* and *Cladosporium fulvum* made *PR-2* and *PR-3* work better together. In the same regard, the antifungal activity of chitinases can be synergistically enhanced by β-1,3-glucanases, both in vitro and in vivo, and these findings are also supported by many studies [56,57], in which the co-expression of chitinase and glucanase genes in tobacco enhanced resistance against *Cercospora nicotianae*. On the contrary, the obtained result demonstrated a decrease in the chitin binding protein (*PR-4*) gene, and this result was a mismatch with that of Hafez et al. [58], where *PR-4* expression was increased in tomato

plants in response to *F. oxysporum* and *R. solani* inoculation; therefore, the plant under *Pythium* inoculation may undergo a defense pathway other than that of the *PR-4* gene.

**Figure 3.** The relative expression level of pathogenesis-related genes (*PR-1*, *PR-2*, *PR-3*, *PR-4*, and *PR-5*) of tomato plants infected with *Pythium aphanidermatum*, collected at different time intervals (5, 10, 15, and 20 days) post inoculation (dpi). Different letters reveal statistically significant variations ($p \leq 0.05$).

Regarding *PR-5* gene expression, the data showed a variable change in *PR-5* (Thaumatin-like proteins, TLPs) gene expression at different sample collection periods. Still, it was highly expressed at 15 dpi and reached 3.99-fold, followed by 20 dpi, where the increasing percentage reached 3.70-fold, relative to the untreated control (Figure 3). The TLPs may have the ability to increase membrane permeability through two possible mechanisms. First, they may cause the destruction of β-1,3-glucans, which are important components of fungal cell walls. Second, they may inhibit fungal enzymes, such as xylanases. This can lead to increased membrane permeability and the subsequent death of the fungus [59]. This finding might support the role of TLPs in the resistance mechanism toward this pathogen, or it may be ascribed to the PR role, which further restricts fungal invasion and its reproduction [9]. Previous studies have indicated that TLPs are induced following fungal infection in plants, including in fruit species. However, there is a significant lack of understanding regarding the genotypic variability of TLPs and their downstream effects. Further research is needed to elucidate the specific roles of TLPs in response to fungal infection in different plant species, particularly in fruits [46,60–62]. Overall, the pathological effects of *P. aphanidermatum* on tomato plants evaluated by the expression of PR genes at different time intervals may require more insights into their role in disease resistance in the future.

*3.4. HPLC Profiling of Tomato Leaves*

Figure 4 displays the HPLC chromatograms of the ethanolic extracts from tomato leaves infected with *P. aphanidermatum* at 20 dpi and those which were uninfected (control). The HPLC analysis indicated that the total concentration of 19 polyphenolic compounds was 3858 µg/g and 3202.2 µg/g in the control and infected plant leaves, respectively, which

means that the leaves showed a low amount of phenolic content after infection. According to Khatri et al. [63], rice leaves infected with *Entyloma oryzae* showed a reduction in the amount of phenols. Naik et al. [64] observed a quick decrease in phenolic compounds in betel vine leaves infected with *Colletotrichum glocosporioides*. Nema [65] believed that phenolic compounds could inhibit the growth of the pathogen, but in cases where the pathogen caused disease, the ratio of phenolic compounds changed. Highly susceptible cultivars were more likely to experience a depletion of phenolic compounds.

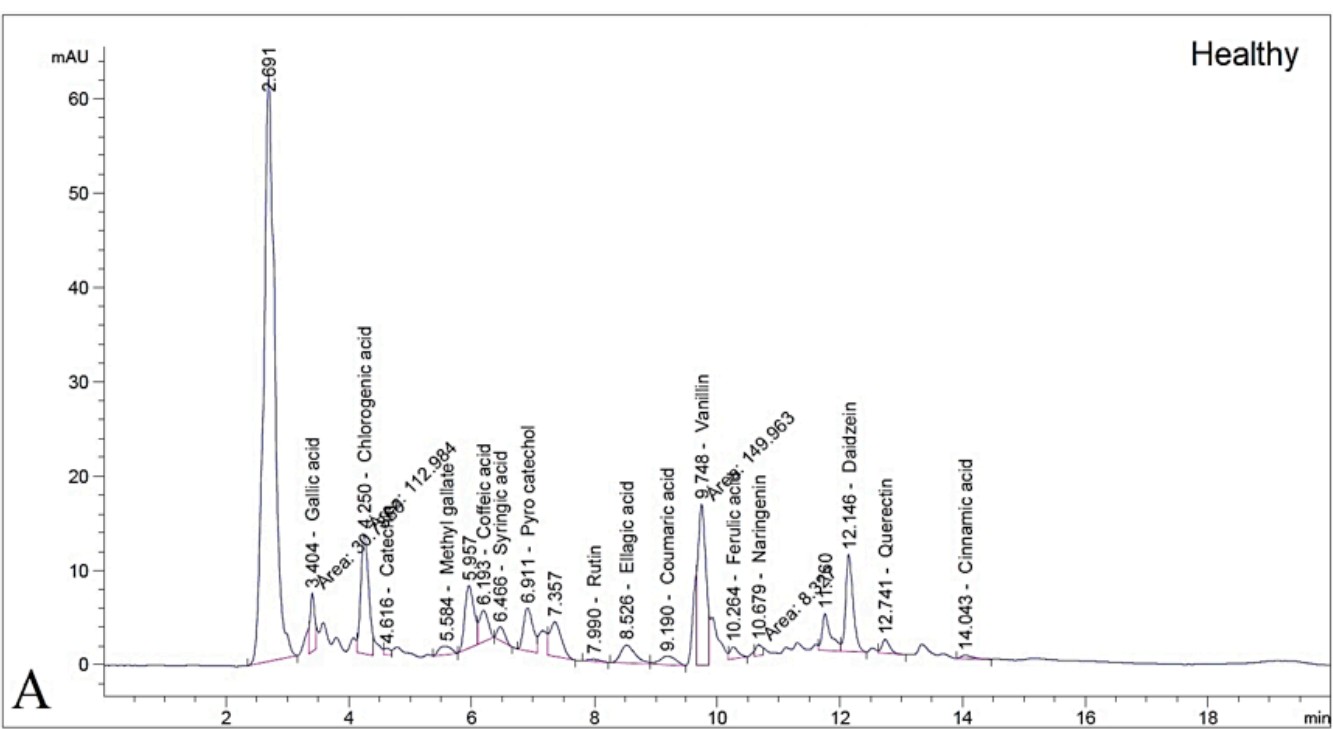

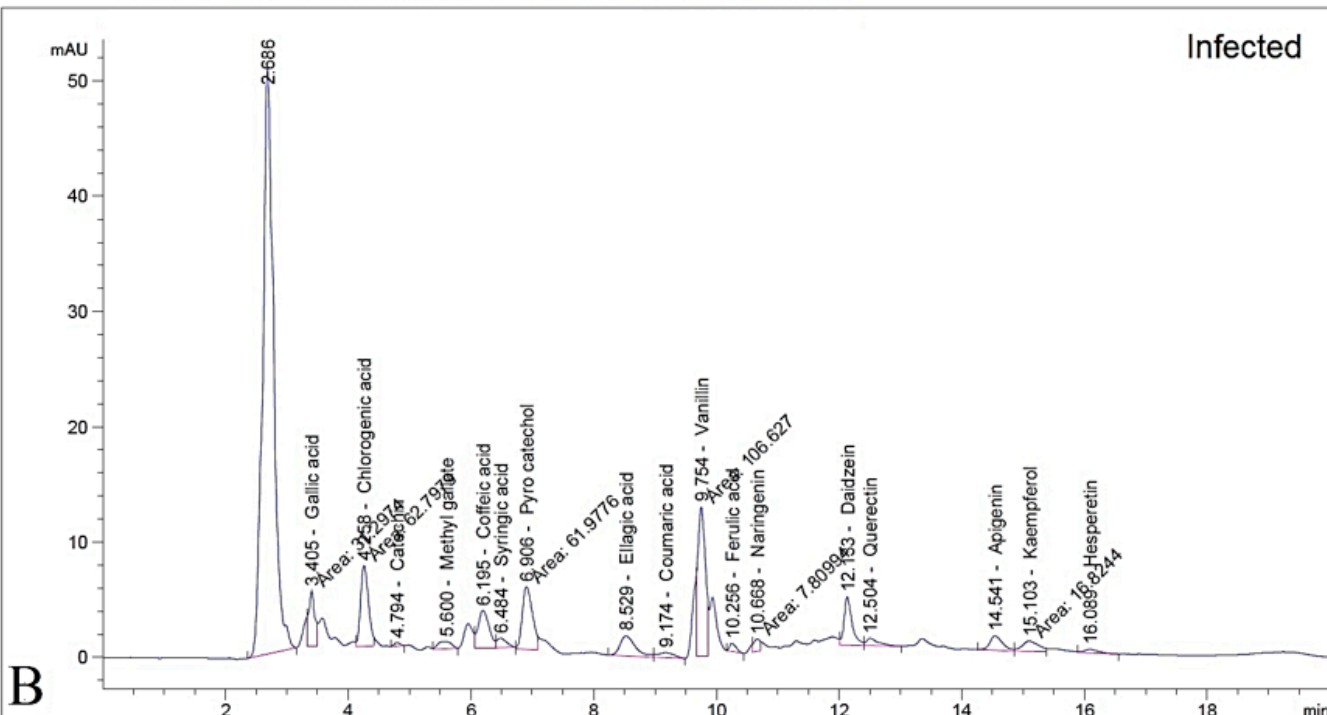

**Figure 4.** (**A**) HPLC profiles of the major polyphenolic compounds in healthy tomato leaves and (**B**) infected tomato plant leaves at 20 dpi.

The main compounds detected are listed in Table 2. The most abundant phenolic compounds in terms of µg/g were chlorogenic acid, ellagic acid, pyrocatechol, and gallic acid in both the control and infected leaf extracts. However, vanillin, a flavonoid compound, was expressed at a lower level in infected plants (299.3 µg/g) than in control plants (421 µg/g) (Table 2). Contrarily, other investigators found that the most abundant flavonoids were vanillin, daidzein, naringenin, and kaempferol [66]. In contrast, Behiry et al. [67] reported that the antifungal activities of the extracted tomato plant leaves were linked to the functional groups and chemical composition of the phenolic compounds, such as caffeic and chlorogenic acids. The same findings were also noticed by several authors who found potent biological activities of identified plant phenolic substances towards various pathogenic microbes [68,69].

**Table 2.** The polyphenolic compounds detected in the control and infected tomato plant leaves.

| Polyphenolic Compounds | Concentration (µg/g) | |
| --- | --- | --- |
| | **Control** | **Infected** |
| Gallic acid | 192.4 | 195.6 |
| Chlorogenic acid | 1110.4 | 617.2 |
| Catechin | 72.4 | 30.7 |
| Methyl gallate | 45.8 | 35.8 |
| Caffeic acid | 182.1 | 219.2 |
| Syringic acid | 62.8 | 45.0 |
| Pyro catechol | 476.7 | 603.3 |
| Rutin | 30.6 | 0.0 |
| Ellagic acid | 609.8 | 578.9 |
| Coumaric acid | 28.6 | 13.7 |
| Vanillin | 421.0 | 299.3 |
| Ferulic acid | 54.5 | 23.3 |
| Naringenin | 57.9 | 54.3 |
| Daidzein | 403.5 | 177.6 |
| Querectin | 102.3 | 67.9 |
| Cinnamic acid | 7.2 | 0.0 |
| Apigenin | 0.0 | 86.8 |
| Kaempferol | 0.0 | 134.1 |
| Hesperetin | 0.0 | 19.5 |

HPLC analysis indicated an increase in pyrocatechol compound content and a decrease in chlorogenic acid in infected plants compared to control plants. Sharifi-Rad et al. [70] noticed that tomato leaves contained more species of flavonoids, which have toxic activities against a broad range of pathogens and pests. Our research shows that when *P. aphanidermatum* is present in tomato plants, several phenolic and flavonoid compounds, such as chlorogenic, syringic, cinnamic, ferulic, ellagic, and coumaric acids, vanillin, catechin, rutin, methyl gallate, naringenin, daidzein, and quercetin, decrease or disappear at 20 dpi. These results suggest that the infection may reduce phenolic acid levels typically produced by infected plant cells. Interestingly, our results also reveal that certain flavonoid compounds (apigenin, kaempferol, and hesperetin) are up-regulated and accumulate in uninfected plants. These findings are supported by previous studies, which have shown that viral infections can promote the expression of flavonoid biosynthesis genes in various plants, including squash, potato, tomato, and grapevine [16,71–73]. These data suggest that flavonoids may affect the physiological responses to fungal infections. Our data somewhat agrees with that of Ortega-Garca et al. [74], who found leaf extracts containing phenolic compounds could protect onions plants from *R. solani* infection.

## 4. Conclusions

Fungi are a major cause of crop losses worldwide; thus, this study focused on the impact of *P. aphanidermatum* on tomato plants. The researchers identified the fungus and then examined the expression of five pathogenesis-related genes (*PR-1*, *PR-2*, *PR-3*, *PR-4*, and *PR-5*) using qPCR. They found that the expression of *PR-1*, *PR-2*, and *PR-5* increased significantly 5 days after inoculation, with the most pronounced increase in *PR-2* and *PR-5* observed at 15 dpi. *PR-1*, *PR-3*, and *PR-5* also showed an increase at 20 dpi, while *PR-4* expression decreased significantly at all time points. In addition, the researchers used HPLC to analyze polyphenolic compounds in the tomato leaves and found that the fungus infection led to a decrease in certain phenolic acids, such as chlorogenic and ellagic acids. Overall, the findings suggest that *P. aphanidermatum* is a biotic stress pathogen that induces the expression of pathogen-related genes and disrupts the regulation of defensin phenolic compounds in tomato plants. Finally, this research can help in the creation of tools and metrics for measuring sustainability that can be used to judge the sustainability of agricultural systems. By determining the degree to which plant diseases affect agricultural productivity, human health, and the environment, researchers can come up with ways to measure the sustainability of farming practices and find ways to make them better.

**Author Contributions:** Conceptualization, S.A.S., A.A.A.-A., S.S., M.A.S., E.H., O.A.S., Y.S., S.I.B. and A.A.; methodology, S.A.S., A.A.A.-A., S.S., M.A.S., E.H., O.A.S., Y.S., S.I.B. and A.A.; software, A.A. and S.I.B.; validation, S.A.S., A.A.A.-A., S.S., M.A.S., E.H., O.A.S., Y.S., S.I.B. and A.A.; formal analysis, S.A.S., A.A.A.-A., S.S., M.A.S., E.H., O.A.S., Y.S., S.I.B. and A.A.; investigation, S.S., M.A.S. and E.H.; resources, S.A.S., A.A.A.-A., S.S., M.A.S., E.H., O.A.S., Y.S., S.I.B. and A.A.; writing—original draft preparation, S.A.S., A.A.A.-A., S.S., M.A.S., E.H., O.A.S., Y.S., S.I.B. and A.A.; writing—review and editing, S.A.S., A.A.A.-A., S.S., M.A.S., E.H., O.A.S., Y.S., S.I.B. and A.A.; supervision, A.A.A.-A. and A.A.; project administration, A.A. and A.A.A.-A.; funding acquisition, A.A.A.-A. All authors have read and agreed to the published version of the manuscript.

**Funding:** This research was financially supported by the Researchers Supporting Project (RSP2023R505), King Saud University, Riyadh, Saudi Arabia.

**Institutional Review Board Statement:** Not applicable.

**Informed Consent Statement:** Not applicable.

**Data Availability Statement:** The data that support the findings of this study are available on request from the corresponding author.

**Acknowledgments:** The authors express their sincere thanks to the City of Scientific Research and Technological Applications (SRTA-City) and the Faculty of Agriculture (Saba Basha), Alexandria University, Egypt, for providing the necessary research facilities. The authors would like to extend their appreciation to the Researchers Supporting Project (RSP2023R505), King Saud University, Riyadh, Saudi Arabia.

**Conflicts of Interest:** The authors declare no conflict of interest.

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
