# Peer review of "Differences in Pathogenesis-Related Protein Expression and Polyphenolic Compound Accumulation Reveal Insights into Tomato–Pythium aphanidermatum Interaction"

_sustainability, doi:10.3390/su15086551_

Round 1
Reviewer 1 Report
Line 23, change ‘which proved’ to ‘which was proved’
Line 25, please provide the full name of ‘qPCR’
Line 32, delete ‘ability to infect tomato plants’
Line 40, delete ‘plant diseases’
Line 42, delete ‘these’
Line 73, change ‘Pythium’ to ‘P.’
Line 93, change ‘other’ to ‘fresh’
Line 112, change ‘a half hour’ to ‘half an hour’
Line 125, change ‘prepared’ to ‘being prepared’
Line 131, change ‘the fungal isolate DNA-ITS region’ to ‘rDNA-ITS region of the fungal isolate’
Line 150, change ‘a minute’ to ‘1 min’
Line 152, change ‘minutes’ to ‘min’. Same as the following.
Line 156, change ‘Gene Bank’ to ‘GenBank’
Line 163, change ‘grams’ to ‘g’. Same as the following.
Line 173, what does the sentence ‘Healthy tomato seedlings served as controls’ mean? The tomato seedlings were not inoculated with Pythium served as controls?
Line 178, change ‘samples, according’ to ‘samples according’
Line 182, change ‘ml’ to ‘mL’. Same as the following.
Line 188, change ‘1 hour’ to ‘1 h’
Line 195, delete ‘reaction’
Line 201, change ‘1.30 hours’ to ‘one and a half hours’
Line 207, change ‘25 25 μL’ to ‘25 μL’
Line 210, change ‘μl’ to ‘μL’. Same as the following.
Line 213, change ‘sec’ to ‘s’. Same as the following.
Line 247, delete ‘genus’
Lines 252-253, change ‘antheridia (1, B); P. aphanidermatum hyphae (2, B); infected tomato plant (C).’ to ‘antheridia (B); P. aphanidermatum hyphae (C); infected tomato plant (D).’
Line 255, change ‘Internal Transcribed Spacer’ to ‘internal transcribed spacer’
Line 271, delete ‘isolate’
Line 272, delete ‘The cluster which found’
Line 307, ‘Nicotiana tabacum’ should be italicized.
Line 326 and Line 328, change ‘Alternaria’ to ‘A.’
Line 334, change ‘Chitin Binding Protein’ to ‘chitin binding protein’
Line 357, change ‘times’ to ‘time’
Line 362, delete ‘plant’
Line 364, change ‘3858’ to ‘3858 μg/g’
Line 391, change ‘days post-inoculation’ to ‘dpi’
Line 394, change ‘research’ to ‘researches’
In Figure 4, change ‘a’ and ‘b’ to ‘A’ and ‘B’, respectively.
Line 407, change ‘Pythium’ to ‘P.’
Line 436, ‘Fusarium’ should be italicized.
Line 438, ‘Pythium aphanidermatum’ should be italicized.
Line 440, change ‘Decomposition by Pythium and Its Relevance to Substrate-Groups of Fungi.’ to ‘decomposition by Pythium and its relevance to substrate-groups of fungi.’
Line 442, ‘Bacillus subtilis’ should be italicized.
Line 446, change ‘Dna’ to ‘DNA’
Reference 7 and Reference 44, please provide the whole information.
Line 455, ‘Pectobacterium carotovorum’ should be italicized.
Lines 463-464, change ‘Bio-Friendly Formulations of Chitinase-Producing Streptomyces cellulosae Actino 48 for Controlling Peanut Soil-Borne Diseases Caused’ to ‘bio-friendly formulations of chitinase-producing Streptomyces cellulosae actino 48 for controlling peanut soil-borne diseases caused’. In addition, ‘Streptomyces cellulosae’ should be italicized.
Lines 468-469, change ‘Defense Responses and Metabolic Changes Involving Phenylpropanoid Pathway and PR Genes in Squash (Cucurbita pepo L.)’ to ‘defense responses and metabolic changes involving phenylpropanoid pathway and PR genes in squash (Cucurbita pepo L.)’. In addition, ‘Cucurbita pepo’ should be italicized.
Line 470, delete ‘(Basel, Switzerland)’
Line 476, ‘Pythium irregular’ should be italicized.
Line 481, what does ‘others’ mean?
Line 494 and Line 512, ‘Phytophthora infestans’ should be italicized.
Lines 497-498, ‘Alternaria brassicicola’ and ‘Arabidopsis thalianaecotype’ should be italicized.
Line 507, ‘Vitis vinifera’ should be italicized.
Line 520, change ‘Lett’ to ‘Lett.’
Line 522, ‘Agave tequilana’ should be italicized.
Line 528, ‘Alternaria solani’ should be italicized.
Line 534, change ‘byCladosporium fulvumand’ to ‘by Cladosporium fulvum and’. In addition, ‘Cladosporium fulvum’ should be italicized.
Line 549, change ‘cell’ to ‘Cell’
Lines 552-553, ‘Malus domestica’ should be italicized.
Line 554, ‘Entyloma oryzae.’ should be italicized.
Lines 556-557, ‘Colletotrichum gloeosporioides’ should be italicized.
Line 561, ‘Trichoderma pubescens’ should be italicized.
Line 562, ‘Rhizoctonia solani’ should be italicized.
Line 566, change ‘plant’ to ‘Plant’
Line 569, ‘Bacillus licheniformis’ should be italicized.
Line 571, ‘Bacillus velezensis’ and ‘Fusarium oxysporum’ should be italicized.
Line 574, ‘Vitis vinifera’ should be italicized.
Line 577, ‘Trichoderma asperellum’ should be italicized.
Line 578, delete ‘(Amsterdam)’
Reviewer 2 Report
This study isolated a pathogenic fungus Pythium aphanidermatum, which caused damping-off disease in tomato plants. This study also quantified the expression level changes of pathogenesis-related genes and concentration changes of phenolic compounds in tomato leaves. This study didn’t describe results clearly and correctly, didn’t provide correct figure legends and didn’t use the same reference style.
It would be helpful to provide a brief title for each figure, set panel names in the same style and put panel names in the beginning of each statement.
It would be helpful to make a table listed the information of 5 pathogenesis-related genes, such as GenBank accession number, uniprot ID, name or function.
1. In line 207, delete one ‘25’;
2. In line 220, there are 5 defense genes and housekeeping β-actin in the table. It would be better to provide a new table title including all information.
3. In line 229-231, what’s meaning of (A) and (B)?
4. In line 251-253, the figure legends didn’t descript the panel clearly. It would be helpful to separate panel B1,2 to different panels, such as B and C. In panel C of figure 1, it would be helpful to display the local infection region.
5. In line 293, the expression level of PR-4 decreased in tomato leaves infected with P. aphanidermatum. Is it correct to use ‘defense genes’ in this title?
6. In line 301, it would be helpful to describe ‘6.34-fold’ clearly in the first time, such as ‘compared to β-actin’.
7. In line 314-315, PR-2 only decreased significantly at 10 dpi and didn’t show up-regulation compared to control.
8. In line 322-337, author should descript content clearly, and it’s better not to use reference number directly.
9. In figure 4, it is possible to show overlay curve of healthy and infected sample results?
10. In line 409-411, PR-2 didn’t increase significantly after inoculation.
11. In line 411-412, PR-4 didn’t decrease significantly after inoculation.
Reviewer 3 Report
This is an excellent contribution to understanding fungal pathogenicity. My opinion is that the MS should be accepted as presented
Author Response
Dear respected reviewer: Many thanks for your support comments concerning our article.
Reviewer 4 Report
Dear,
I read carefully this manuscript and I think, it can be considered for publication. Authors investigated a fungus isolate was isolated from infected tomato plants and molecularly identified as Pythium aphanidermatum (deposited in GenBank under accession number MW725032), which proved to be pathogenic and caused damping-off disease. There are some comments which should be considered before next step:
-Abstract is not suitable and should be revised and explain more the results part.
-In introduction section: the paragraph (Plant polyphenolic chemicals, like phenolic .......) (Line 67-72) should be more explained and I suggested you the below references;
Barzegar, P.E.F., et al., The current natural/chemical materials and innovative technologies in periodontal diseases therapy and regeneration: A narrative review. Materials Today Communications, 2022. 32: p. 104099.
-Method and materials are Ok
-Results and discussion sections are poor, author should use the recent studies and compare their results with other studies.
Best,
Round 2
Reviewer 2 Report
The authors have answered all my questions clearly. There are only 2 comments left:
1. In line 178, it’s better to italicize ‘Pythium’.
2. In line 314, the temperatures should be ’27 ℃’ and ’20 ℃’. It would be helpful to delete ‘-‘.
Author Response
Thanks for your comments
We checked and corrected the two issues throughout the manuscript
Reviewer 4 Report
Dear,
The revised manuscript is acceptable.
Best,
Author Response
Many thanks for your support